# Associations between neighbourhood characteristics and participation in a population-based organised prostate cancer testing (OPT) programme: A register-based study of 50-year-old men

Emil Järbur[1,2]*, Rebecka Arnsrud Godtman[1,3], Carl Bonander[4], Ulf Strömberg[4], Ola Bratt[1,3]

1 Department of Urology, Institute of clinical sciences, Sahlgrenska academy, University of Gothenburg, Gothenburg, Sweden, 2 Department of Urology, Uddevalla hospital, Uddevalla, Sweden, 3 Department of Urology, Sahlgrenska University Hospital, Gothenburg, Sweden, 4 School of public health and community medicine, Institute of Medicine, University of Gothenburg, Gothenburg, Sweden

* emil.jarbur@vgregion.se, emil.jarbur@gmail.com

## Abstract

### Objectives

Regional, population-based organised prostate cancer testing (OPT) began in Sweden in 2020. We investigated associations between participation and neighbourhood characteristics.

### Setting

Region Västra Götaland's OPT programme

### Methods

Data were retrieved from the regional OPT database, for all 50-year-old men invited in 2020–2021. Addresses were linked to demographic statistical areas defined and socioeconomically described by Statistics Sweden. Logistic regression models were used to analyse participation based on neighbourhood deprivation, income, education, proportion of non-western immigrants, and degree of urbanisation, categorised in quintiles. Results are reported as unadjusted and adjusted odds ratios (ORs) with confidence intervals (CIs).

### Results

Unadjusted participation was significantly lower in the quintiles of neighbourhoods with the highest deprivation index (OR 0.61, 95% CI 0.56–0.67), lowest income (OR 0.64, 95% CI 0.59–0.71), lowest education (OR 0.81, 95% CI 0.74–0.89), and highest proportion of non-Western immigrants (OR 0.70, 95% CI 0.64–0.77), compared with

**Data availability statement:** All data generated or analysed during this study are included in this published article (and its supplementary information files).

**Funding:** Västra Götalandsregionen, Swedish Research Council for Health, Working life and Welfare (Forte) Grant/Award number: ALFGBG-965228 OB, VGFOUREG-993861 OB, Forte-2023-01104 US and CB The funders had no role in study design, data collection and analysis, decision to publish, or preparation of the manuscript.

**Competing interests:** Emil Järbur certifies that all conflicts of interest, including specific financial interests and relationships and affiliations relevant to the subject matter or materials discussed in the manuscript, are the following: Rebecka Arnsrud Godtman has received lecture fees and travel honoraria from Bayer and Ipsen. Emil Järbur, Ulf Bonander, Ulf Strömberg and Ola Bratt have no conflicts of interest other than working with and/or doing research related to OPT. This does not alter our adherence to PLOS ONE policies on sharing data and materials.

the opposite quintile. After adjustment for the other variables, significant gradients remained for deprivation (OR 0.65, 95% CI 0.57–0.74), proportion of non-Western immigrants (OR 0.78, 95% CI 0.69–0.88), and degree of urbanization (OR 0.78, 95% CI 0.71–0.87 for rural versus urban areas). No adjusted analysis was done for income owing to its strong correlation with deprivation.

## Conclusion

Socioeconomic factors and degree of urbanisation influence participation in organised screening for prostate cancer. Actively inviting men for screening, does not in itself avoid socioeconomic disparity in early detection of prostate cancer.

## Introduction

Population-based cancer screening programmes aim at reducing morbidity and mortality. A secondary aim may be to reduce socioeconomic inequity in early detection of cancer. However, research has shown that the introduction of a screening programme may exacerbate rather than reduce inequality [1,2]. Socioeconomic factors and degree of urbanisation affect participation in established screening programmes such as breast, cervical, and colorectal cancer screening [3–7]. A first step in efforts to reduce inequality is to apply measures for monitoring participation rates across various socioeconomical groups; measures that can subsequently be used for the evaluation of the effectiveness of interventions to improve participation in specific population subgroups.

Prostate-specific antigen (PSA) based screening has been shown to reduce prostate cancer mortality but at the price of considerable overdiagnosis and overtreatment [8,9]. Novel diagnostic pathways with multiparametric magnetic resonance imaging (MRI), PSA density and lesion-targeting biopsies have substantially reduced overdiagnosis [10–13]. In 2018, the Swedish Board of Health and Welfare recommended against a national PSA-based screening programme but encouraged regional initiatives to organise the widespread opportunistic PSA testing [14]. This led to implementation of regional, population-based, organised prostate cancer testing (OPT) programmes, the first two of which were started in 2020 [15,16]. As of November 2024, 14 of the 21 regions in Sweden have started OPT. In essence, the OPT programmes are structured similarly to the established Swedish cancer screening programmes, but the invitation includes a statement that the national healthcare authorities do not recommend a national screening programme because of uncertainty about whether the benefits outweigh the harms on a population level.

One of the motives for implementing regional OPT programmes is to reduce socioeconomic inequality in prostate cancer diagnostics, another is to gather organisational experience in preparation for a future national screening programme [17]. The Swedish OPT programmes align well with the European Union Council's recommendation in December 2022 to evaluate the feasibility of screening for prostate cancer and then stepwise introduce population-based screening [18].

Participation in organised screening programmes may differ notably between residential neighbourhoods [19,20]. Lower participation in established cancer screening programmes and in prostate cancer screening trials has been reported in deprived and rural areas [19–23], but never from population-based screening for prostate cancer integrated in public healthcare. Moreover, low individual income and low education were associated with a more severe disease at presentation and higher prostate cancer-specific mortality, in both the control and the screening group of a Finnish prostate cancer screening trial [21].

We have previously reported on associations between individual level socioeconomic factors and participation in a regional OPT programme in Sweden [16]. Another method to investigate socioeconomic associations with participation is using neighbourhood characteristics. This method, with usage of anonymous group-level data rather than sensitive individual-level data, is internationally more widely accessible, is more cost-effective and thereby allows for frequent outcome monitoring, and simplifies assessment of the influences of urbanisation and other geographical factors. The aim of this study was to investigate associations between neighbourhood characteristics and participation in a population-based, organised screening programme for prostate cancer.

## Method and materials

### The Region Västra Götaland organised prostate cancer testing (OPT) programme

Sweden's healthcare system is operated by 21 regions. Swedish OPT is in many aspects like regional screening programmes. Men with a prostate cancer diagnosis are identified in the Cancer Register and excluded from the programmes. The invited men receive a letter with nationally standardised, neutral information about the advantages and disadvantages of testing for prostate cancer [24], together with an invitation to participate in the programme by having a PSA test. More in-depth information and a video with sound and sight interpretation are available online.

In 2020, Region Skåne and Region Västra Götaland were the first to start OPT [15,25]. So far, OPT in Region Västra Götaland has invited all men in specific birth cohorts, starting at the age of 50 years. The information letter was at the time of this study available in Swedish, English, Dari, Arabic, Persian and Somali. The men were offered a PSA test within 1 month but could be tested later after contact with the OPT office. The letter included a list of 36 (initially 27) testing facilities at hospitals and primary care centres, spread across the region to minimize the geographic distance to a test facility. The distance to the nearest test facility was usually shorter than 5 km and never longer than 25 km. PSA testing was free of charge for the participants, as well as any subsequent MRI and urological investigation. PSA samples obtained within the OPT programme were specifically labelled to link them with the programme. The public was informed about the OPT programme in local media, such as newspapers and posters at bus stops, as well as in social media during the observed period. Information about invitations and diagnostic data are registered in a regional OPT database.

### Study population

All men invited in the first two years (2020 and 2021) of Region Västra Götaland's OPT programme were included after ethical approval from the Swedish Ethical Review Authority (2020–07098). The men invited in 2020 were born in 1970 and the men invited in 2021 were born 1971. Because of the decision to start with 50-year-old men and the biannual invitation system of the OPT, only 50-year-old men were invited during this period.

### Demographic statistical areas and data collection

In 2018, Statistics Sweden divided the country into 5,985 Demographic Statistical Areas (DeSOs by Swedish acronym), with the aim to facilitate monitoring of segregation and socioeconomic conditions in small areas. By the end of 2021 they had between 447 and 7842 inhabitants. The DeSOs' borders are changed only if the population within a DeSO changes significantly. Since the launch of the DeSOs in 2018 only one change was made (two DeSOs were merged). Statistics Sweden provides comprehensive sociodemographic data for each DeSO.

Postal addresses for individuals receiving invitation letters to the regional OPT programme were collected from the Population Register on 19th September 2022. Each address was assigned (geo-coded) to one of 992 regional DeSOs. Information about PSA testing was obtained from the regional OPT database. Only PSA tests labelled OPT were included. The linking of participation with DeSO was done by Regional Cancer Center West. The authors of this paper did not have access to information that could identify individual participants during or after data collection.

### Neighbourhood characteristics

The proportion of men who participated by having an OPT PSA test was calculated for DeSOs grouped in national quintiles for specific socioeconomic characteristics: Deprivation index, proportion of inhabitants with low economic standard, proportion of inhabitants aged 25–65 years without tertiary education (i.e., up to 12 years of education), proportion of inhabitants born in a non-Western country, and degree of urbanisation [26]. The deprivation index consisted of four factors: proportion of inhabitants with low economic standard, proportion of inhabitants aged 25–65 years without tertiary education, proportion of inhabitants aged 25–65 years without paid employment, and proportion of inhabitants living in a rented accommodation. Economic standard was analysed by disposable income per consumption unit. Disposable income per consumption unit is a metric commonly employed by Statistics Sweden to assess economic standards, representing the total household income available after taxes, adjusted for the number of individuals in the household. Low economic standard was defined as the lower quartile of disposable income per consumption unit among all Swedish households (cutoff at 20,693 USD/year in 2020). No tertiary education was defined as inhabitants aged between 24–65 years with ≤12 years in school (corresponding to a finished secondary education in Sweden or shorter). Non-Western countries were defined as Eastern Europe, Asia, Africa and South America. The DeSOs were categorised as rural, semi-urban or urban by Statistics Sweden.

### Statistical analysis

The outcome measure was participation in the OPT programme, defined as having had an OPT PSA test. DeSO characteristics were categorised in quintiles. The associations between each neighbourhood characteristic and the proportion of participating men were analysed with logistic regression and reported as odds ratios (ORs) with 95% confidence intervals (CIs), crude (unadjusted) and after adjustment for the other analysed factors in multivariable logistic regression models. Correlation between neighbourhood characteristics was explored using cross tabulation shown in supplementary tables 1–4 in S1 Text. We also analysed the population distribution across neighbourhoods in supplementary tables 5 – 9 in S1 Text.

Additionally, we employed a Bayesian model where spatially structured random effects were modelled using an intrinsic conditional autoregressive prior [27,28]. Thereby, we allowed for participation rates to be smoothed across neighbouring DeSOs. The supplements contain a detailed description of this statistical modelling approach. The results from the spatially structured random effect model without covariates (i.e., without adjustments for neighbourhood characteristics) were visualised in maps. We also included the DeSO-level covariates in a spatially structured random effects model and compared the association estimates provided with the corresponding ORs (95% CIs) obtained from the conventional logistic regression model. There were no differences in results between the models, so we show results only for the spatially structured random effect model without covariates.

## Results

A total of 21,176 men aged 50 years were invited during the first two years of OPT (2020–2021). Of these, 20,948 (99%) could be assigned to a DeSO and were included in the analysis. A total of 7,902 (38%) men chose to participate, i.e., had a PSA test within the programme. The proportion of the examined population in the national quintiles for socioeconomic characteristics as well as the proportion of invited men by degree of urbanisation is shown in Table 1.

 

**Table 1. Proportion of invited men in each quintile for examined neighbourhood characteristics and for degree of urbanisation among 20,948 50-year-old men invited in 2020 and 2021 to an organised prostate cancer testing programme in Region Västra Götaland, Sweden.**

| Neighbourhood characteristic | | Proportion of invited men in each quintile (no. invited) |
|---|---|---|
| Deprivation (an index of multiple deprivation)[a] | | |
| | Q1 (least) | 27.0% (5654) |
| | Q2 | 19.3% (4035) |
| | Q3 | 18.0% (3770) |
| | Q4 | 18.0% (3773) |
| | Q5 (most) | 17.7% (3716) |
| Proportion of inhabitants with low economic standard[a] | | |
| | Q1 (lowest) | 26.6% (5577) |
| | Q2 | 20.8% (4363) |
| | Q3 | 19.9% (4171) |
| | Q4 | 17.1% (3577) |
| | Q5 (highest) | 15.6% (3260) |
| Proportion of inhabitants aged 25–64 years with ≤12 years in school[a] | | |
| | Q1 (lowest) | 20.7% (4339) |
| | Q2 | 22.3% (4775) |
| | Q3 | 21.0% (4390) |
| | Q4 | 19.8% (4140) |
| | Q5 (highest) | 15.8% (3304) |
| Proportion of non-Western immigrants[a] | | |
| | Q1 (lowest) | 16.1% (3364) |
| | Q2 | 23.1% (4830) |
| | Q3 | 22.4% (4697) |
| | Q4 | 19.1% (4003) |
| | Q5 (highest) | 19.4% (4054) |
| Degree of urbanisation | | |
| | Urban area | 75.4% (15799) |
| | Semi-urban area | 6.9% (1446) |
| | Rural area | 17.7% (3703) |

[a]Neighbourhood-level data divided into national quintiles (Q) Q1-Q5 for DeSOs in Sweden.

## Participation by socioeconomic context

The OPT participation rates differed across DeSO quintiles for all analysed factors, from between 31% (deprivation index) and 34% (education) in the quintiles with most deprivation and highest proportion of inhabitants aged 25–65 years without tertiary education to between 40% (country of birth) and 43% (deprivation index) in the quintiles with lowest proportion of non-western immigrants and least deprivation (Table 2). The crude participation and unadjusted odds ratio gradients were greatest for deprivation index and income, intermediate for education and country of birth, and smallest for degree of urbanisation (Table 2). Proportion of inhabitants with low economic standard was not included in the adjusted analysis owing to its strong correlation with deprivation index (Spearman's correlation = 0.94). Correlations between DeSO characteristics are shown in supplementary tables 1-4 in S1 Text).

Deprivation index and proportion of inhabitants with low economic standard were the socioeconomic factors that showed the strongest associations with participation in the programme. Using the socioeconomically most favoured quintile as the reference, the OR for participation was 0.61 (95% CI 0.56–0.67) in the quintile with most deprived households

**Table 2. Associations of neighbourhood characteristics with PSA testing among 20,948 50-year-old men invited in 2020 and 2021 to an organised prostate cancer testing programme in Region Västra Götaland, Sweden.**

| Neighbourhood characteristic | | Attendance proportion (no. attendees/no. invited) | Unadjusted OR (95% CI)[a] | Adjusted OR (95% CI)[b] |
|---|---|---|---|---|
| Deprivation (an index of multiple deprivation)[c] | | | | |
| | Q1 (least) | 42·7% (2413/5654) | 1 | 1 |
| | Q2 | 38·8% (1564/4035) | 0·85 (0·78–0·92) | 0·90 (0·82–0·99) |
| | Q3 | 37·2% (1404/3770) | 0·80 (0·73–0·87) | 0·83 (0·76–0·92) |
| | Q4 | 35·9% (1355/3773) | 0·75 (0·69–0·82) | 0·76 (0·69–0·84) |
| | Q5 (most) | 31·4% (1166/3716) | 0·61 (0·56–0·67) | 0·65 (0·57–0·74) |
| Proportion of inhabitants with low economic standard[c] | | | | |
| | Q1 (lowest) | 42·0% (2342/5577) | 1 | (not included[d]) |
| | Q2 | 39·7% (1732/4363) | 0·91 (0·84–0·99) | – |
| | Q3 | 36·6% (1526/4171) | 0·80 (0·73–0·86) | – |
| | Q4 | 35·4% (1265/3577) | 0·76 (0·69–0·82) | – |
| | Q5 (highest) | 31·8% (1037/3260) | 0·64 (0·59–0·71) | – |
| Proportion of inhabitants aged 25–64 years with ≤12 years in school[c] | | | | |
| | Q1 (lowest) | 39·2% (1703/4339) | 1 | 1 |
| | Q2 | 40·9% (1955/4775) | 1·07 (0·99–1·17) | 1·11 (1·02–1·21) |
| | Q3 | 36·5% (1603/4390) | 0·89 (0·82–0·97) | 1·07 (0·97–1·18) |
| | Q4 | 36·4% (1506/4140) | 0·88 (0·81–0·97) | 1·17 (1·05–1·30) |
| | Q5 (highest) | 34·4% (1135/3304) | 0·81 (0·74–0·89) | 1·18 (1·04–1·33) |
| Proportion of non-Western immigrants[c] | | | | |
| | Q1 (lowest) | 39·9% (1343/3364) | 1 | 1 |
| | Q2 | 39·7% (1919/4830) | 0·99 (0·91–1·09) | 0·96 (0·87–1·05) |
| | Q3 | 39·8% (1869/4697) | 0·99 (0·91–1·09) | 0·99 (0·90–1·09) |
| | Q4 | 37·0% (1480/4003) | 0·88 (0·80–0·97) | 0·91 (0·82–1·01) |
| | Q5 (highest) | 31·8% (1291/4054) | 0·70 (0·64–0·77) | 0·78 (0·69–0·88) |
| Degree of urbanisation | | | | |
| | Urban area | 38·2% (6038/15799) | 1 | 1 |
| | Semi-urban area | 35·2% (509/1446) | 0·88 (0·78–0·98) | 0·80 (0·71–0·90) |
| | Rural area | 36·6% (1355/3703) | 0·93 (0·87–1·00) | 0·78 (0·71–0·87) |

[a]Odds ratios (ORs) with 95% confidence interval (CI) obtained from logistic regression including a single neighbourhood-level covariate.

[b]Odds ratios (ORs) with 95% confidence interval (CI) obtained from the multivariable logistic regression model obtained after a statistical selection procedure.

[c]Neighbourhood-level data divided into national quintiles (Q) Q1-Q5 for DeSOs in Sweden.

[d]Proportion of inhabitants with low economic standard was not included because of its strong correlation with deprivation index.

and 0.64 (95% CI 0.59–0.71) in the quintile with the highest proportion of inhabitants with low economic standard. The participation rates gradually increased for each quintile nearer to the reference (Table 2). The analysis of the proportion of inhabitants aged 25–64 years with ≤12 years in school and the proportion of non-Western immigrants both showed significant differences in the unadjusted ORs, but on adjusted analysis the association was reduced for the proportion of non-Western immigrants and disappeared for education (Table 2).

**Participation by degree of urbanisation**

The unadjusted odds ratios for participation by degree of urbanisation were of borderline statistical significance, but after adjustment for deprivation, education and proportion of immigrants, the ORs became significant: rural 0.78 (95% CI 0.71–0.87) and semi-urban 0.80 (95% CI 0.71–0.90) versus urban areas (Table 2).

## Participation by mapping of neighbourhoods

The maps generated from the spatially structured random effect model are shown in Fig 1, with one map visualising the estimated participation rates by shifted colours from light green (lowest) to dark blue (highest) and the other map showing statistical signals for lowered participation compared to the average participation rate in the whole region (Fig 2).

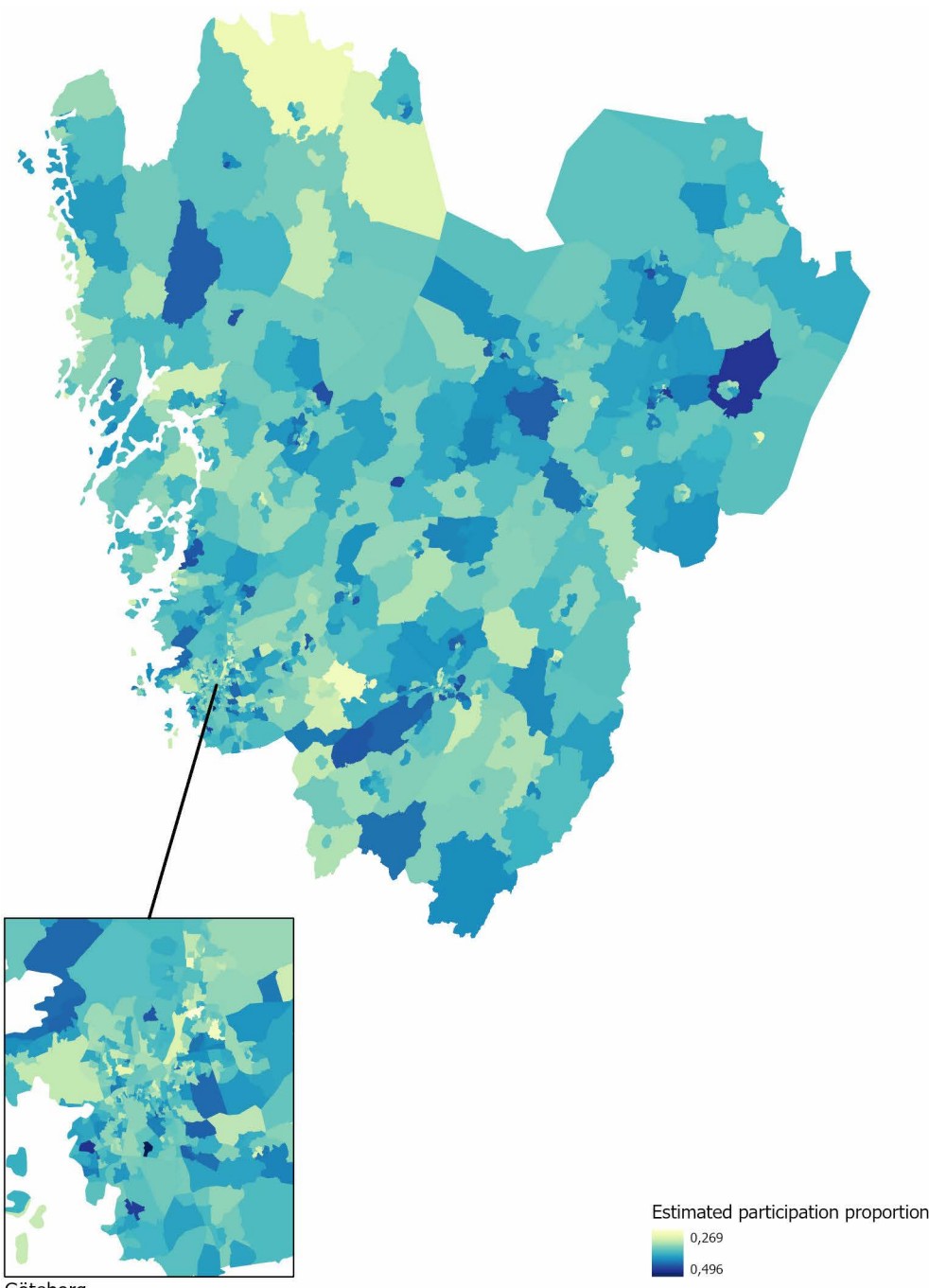

Göteborg

**Fig 1. A map showing Region Västra Götaland, with the Gothenburg urban area magnified, and estimated participation rate in the organised testing programme for prostate cancer in 2020–2021.** Shifted colours from light green (lowest participation) to dark blue (highest).

Five neighbourhoods (DeSO) showed strong statistical signals for reduced participation, with estimated participation proportion below 30% (Fig 2). Three of these DeSOs were in urban areas, classified in the most deprived national quintile (Q5) and, also, in the highest proportion of non-western immigrant quintile (Q5). The other two were rural, with different

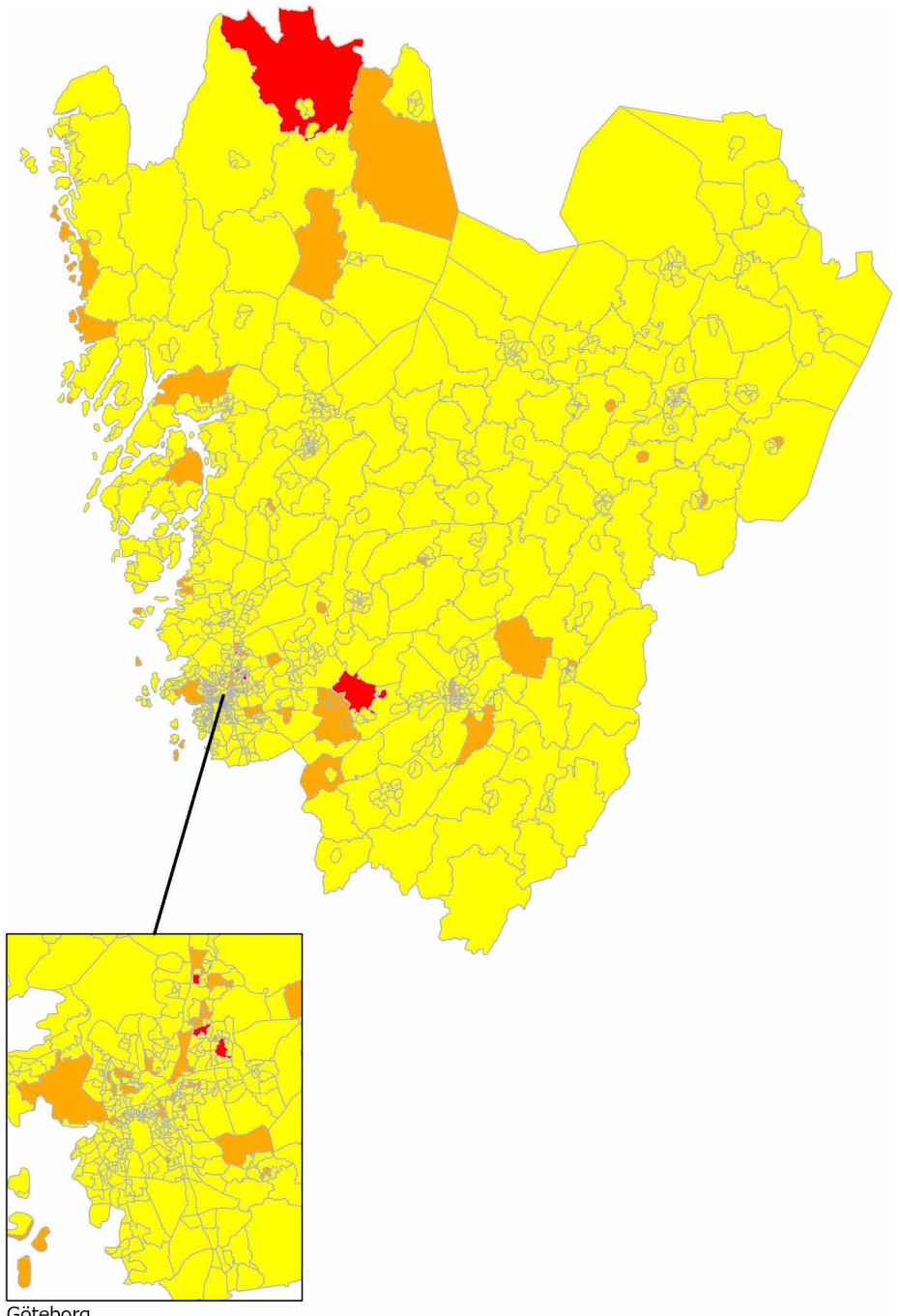

Göteborg

**Fig 2. A map showing Region Västra Götaland, with the Gothenburg urban area magnified, and statistical evidence for reduced participation rates compared to the mean participation in the organised testing programme for prostate cancer in 2020–2021.** Red areas have strong statistical evidence for reduced participation rates and orange areas have significant evidence for reduced participation rates. (red ≥ 95% posterior probability of lowered participation, orange = 80-95% posterior probability, and yellow = < 80% posterior probability) [29].

sociodemographic characteristics: one DeSO was classified in the least deprived quintile (Q1) and the lowest proportion of non-western immigrant quintile (Q1), while the other was classified as more deprived (Q4) yet in the lowest proportion of non-western immigrant quintile (Q1).

## Discussion

This is the first report on associations between socioeconomic neighbourhood characteristics and participation in a population-based, organised prostate cancer testing programme. Our analysis revealed moderate differences in participation rates across neighbourhood characteristics for all analysed socioeconomic factors. The participation gradient was greatest for deprivation index: from 31% to 43%; the gradients related to income, education and proportion on non-Western immigrants were slightly smaller, but education was associated with participation on unadjusted analysis only. As income was strongly correlated with deprivation index, our results suggest that, in Sweden, neighbourhood income is more strongly associated with OPT participation than neighbourhood education level and proportion of non-Western immigrants in the neighbourhood. This agrees well with a report from the Swedish cervical cancer screening programme [30] as well as with a report of single deprivation indicators versus an index of deprivation on all cause mortality [26]. Significant differences by degree of urbanisation were observed only after adjustment for the other factors, when participation was lower in rural and semi-urban than in urban areas (ORs 0.78 and 0.80).

Observed socioeconomic inequality in screening programmes should lead to interventions aimed at reducing disparity. While many such interventions have been explored, evidence for their effectiveness remains limited [31, 32]. Reminders and increasing accessibility, such as providing home-based testing options, have proven effective in some screening contexts [7,33,34]. Home-based testing has proven effective in cervical cancer prevention [33, 34], and home-based PSA testing is currently being evaluated in Swedish regions with a high proportion of men with a long distance to a testing facility. Interventions that are effective for raising participation in one screening programme may, however, not work in others.

### Comparisons with established cancer screening programmes

Socioeconomic participation gradients have been previously documented in established screening programmes, both Sweden and elsewhere [6,35,36]. The associations found in the initial phase of the regional OPT programme that we investigated are consistent with these reports, despite a lower overall participation rate.

Sweden has a long history of high participation in national screening programmes [37,38]. The Swedish population's high trust in authorities often leads to compliance with their recommendations [39,40]. The participation rate in the OPT (38%) was notably lower than that in the established national screening programmes for breast cancer (81%) and cervical cancer (79%) [41]. A key reason for this difference may be the lack of a recommendation for prostate cancer screening from the Swedish health authorities. The invitation to OPT includes neutral information that outlines potential harms as well as benefits, without any directive to participate or a date and time to have a PSA test, and no reminder was sent to non-participants. This contrasts with other Swedish screening programs, to which participation is recommended by health authorities. The newness of the OPT concept may contribute to a lower participation rate.

The relatively young age of the invited men may also have influenced participation rates. Younger men are less likely to obtain PSA testing [42–44], and younger UK men are less likely than older men to participate in colorectal cancer screening [32]. Furthermore, men may be less likely than women to participate in cancer screening [45].

### Comparison with studies using individual level data

We recently reported on the associations between socioeconomic factors and participation in the OPT, using individual-level data from the same population [16]. Just as in the present report, based on neighbourhood characteristics, the individual level analysis found that income had the strongest association with participation. However, in that analysis, significant associations were found for all examined factors: income, education, cohabitation status, and country of birth.

Despite differences in the definitions of low economic status, both studies found strong associations between low income and lower participation rates, with an incidence rate ratio of 0.52 in the individual-level data study and an unadjusted odds ratio of 0.64 in this study. Even though the PSA testing was without cost in the OPT programme, low income was the contextual factor with the greatest impact on participation rates.

Education showed the weakest association with participation in the individual-level study, and no association was found on multivariable analysis in this report based on statistical areas. When making comparisons between analyses based on neighbourhood and individual-level socioeconomic data, it is important to consider the inherent methodological differences. In the individual-level study, education was categorized as primary (<11 years), secondary (11–12 years), and tertiary (>12 years), with comparisons made to secondary education as the reference as this was most common. In this report, education was analysed by proportion of inhabitants aged 25–64 with ≤12 years of education (primary and secondary together). Among the 50-year-old men in the studies, 62% had completed ≤12 years of education. The analysis of the proportion of inhabitants aged 24–65 years with ≤12 years of education is not an assessment of low education but rather of non-tertiary versus tertiary education. Our decision to focus on the selected characteristic was guided by its use in previous studies utilising DeSOs [26]. To investigate the role of low education, a different characteristic, such as analysing the proportion of inhabitants aged 24–65 years with <11 years of education can be used within the DeSO framework.

Another important methodological difference concerns the analyses of country of birth. Individual-level data assesses the influence of personal factors such as cultural background or language barriers, while neighbourhood-level data focuses on contextual factors and their associations with the cultural environment. While the individual-level study categorised countries as Nordic or non-Nordic [16], this report used the proportion of non-Western immigrants. Differences between non-Nordic and non-Western countries of birth may be relevant to participation rates. We used the proportion of non-Western immigrants as this is the most common definition in DeSO-based analyses. In our previous study, we used non-Nordic because we believed that was the most important distinction to detect differences in participation.

### Implications

Using neighbourhood characteristics to examine associations between socioeconomic and geographical factors and participation in population-based screening is a practical approach for monitoring trends over time. This method is more cost-effective, time-efficient, and less resource-intensive than using individual-level data, making it suitable for frequent monitoring. While it cannot replace individual-level data for investigating the impact of individual factors on participation, it may be the most feasible way to track socioeconomic associations with screening participation. Work is ongoing to integrate neighbourhood (DeSO) codes in the regional and national OPT registers to facilitate routine, annual evaluation of socioeconomic inequalities.

This approach can also be applied to design and evaluate neighbourhood level interventions aimed at improving participation rates related to socioeconomic or geographic factors [3,46]. The participation map (Fig 2) can identify areas with low participation rates that are not associated with specific socioeconomic characteristics and inform geographically targeted efforts to increase participation.

### Strengths, weaknesses and limitations

Our study was based on population-level data from reliable national registers. The dataset was large with minimal missing data and there was thereby a negligible risk of bias.

The external validity of our results may be limited to countries with socioeconomic conditions like Sweden's. Sweden is ranked 20 of 161 countries in the Commitment to Reducing Inequality Index [47] and has low socioeconomic gaps compared to many other European countries – although the gaps are growing [48]. The inequalities reported here are likely greater in countries with less subsidised healthcare or for screening programs to which participation is not free of cost. Another limitation is that the study included only men aged 50 years. The influence of socioeconomic factors on participation may be different in older men.

## Conclusions

Men residing in neighbourhoods characterised by high deprivation, low income, high proportion of non-Western immigrants, and non-urban settings are less likely to participate in organised prostate cancer testing in Sweden. By utilising neighbourhood characteristics, we can analyse socioeconomic factors influencing participation and create maps to target specific communities with lower attendance. These findings agree well with reports from established cancer screening programmes. Further research is necessary to explore the underlying reasons for socioeconomic disparities and to identify interventions that ameliorate these inequalities. Our findings are particularly relevant following the European Union's recommendation to stepwise implement prostate cancer screening. Actively inviting men to screening does not eliminate socioeconomic disparities in the early detection of prostate cancer. A crucial first step towards promoting equal participation in organised testing programmes is to assess existing disparities and develop strategies to mitigate them.

## Supporting information

**S1 Text. Supplemetary equation, tables and text.** Text file containing the equation for spatial modelling, supplementary tables 1–9 and a text about population distribution across neighbourhoods.
(DOCX)

## Acknowledgments

Ann Carlstrand and Sigrid Bengtsson at Region Västra Götaland's OPT office for providing information about various organisational details.

## Author contributions

**Conceptualization:** Emil Järbur, Rebecka Arnsrud Godtman, Ulf Strömberg, Ola Bratt.

**Data curation:** Emil Järbur, Rebecka Arnsrud Godtman, Ulf Strömberg.

**Formal analysis:** Ulf Strömberg, Ola Bratt.

**Funding acquisition:** Ola Bratt.

**Investigation:** Emil Järbur, Rebecka Arnsrud Godtman, Carl Bonander, Ola Bratt.

**Methodology:** Emil Järbur, Rebecka Arnsrud Godtman, Carl Bonander, Ulf Strömberg, Ola Bratt.

**Project administration:** Emil Järbur, Rebecka Arnsrud Godtman, Ola Bratt.

**Supervision:** Rebecka Arnsrud Godtman, Carl Bonander, Ulf Strömberg, Ola Bratt.

**Validation:** Ulf Strömberg, Ola Bratt.

**Visualization:** Emil Järbur, Carl Bonander, Ulf Strömberg.

**Writing – original draft:** Emil Järbur, Ola Bratt.

**Writing – review & editing:** Rebecka Arnsrud Godtman, Carl Bonander, Ulf Strömberg, Ola Bratt.

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
