## [Decision Letter · Decision Letter 0]

13 Feb 2025

PONE-D-25-02150Associations between neighbourhood characteristics and participation in a population-based organised prostate cancer testing (OPT) programme: a register-based study of 50-year-old menPLOS ONE

Dear Dr. Järbur,

Thank you for submitting your manuscript to PLOS ONE. After careful consideration, we feel that it has merit but does not fully meet PLOS ONE’s publication criteria as it currently stands. Therefore, we invite you to submit a revised version of the manuscript that addresses the points raised during the review process.

Please respond to all the comments of the two reviewers and if applicable justify why you do not think it is appropriate to edit the text as they suggest.

We look forward to receiving your revised manuscript.

Kind regards,

Lorenzo Righi

Academic Editor

PLOS ONE

“Funder: Västra Götalandsregionen, Swedish Research Council for Health, Working life and Welfare (Forte)

Grant/Award number: ALFGBG-965228 OB, VGFOUREG-993861 OB, Forte-2023-01104 US and CB”

“Emil Järbur certifies that all conflicts of interest, including specific financial interests and relationships and affiliations relevant to the subject matter or materials discussed in the manuscript, are the following:

Rebecka Arnsrud Godtman has received lecture fees and travel honoraria from Bayer and Ipsen.

Emil Järbur, Ulf Bonander, Ulf Strömberg and Ola Bratt have no conflicts of interest other than working with and/or doing research related to OPT.”

5. We note that Figures 1 and 2 in your submission contain [map/satellite] images which may be copyrighted. All PLOS content is published under the Creative Commons Attribution License (CC BY 4.0), which means that the manuscript, images, and Supporting Information files will be freely available online, and any third party is permitted to access, download, copy, distribute, and use these materials in any way, even commercially, with proper attribution. For these reasons, we cannot publish previously copyrighted maps or satellite images created using proprietary data, such as Google software (Google Maps, Street View, and Earth). For more information, see our copyright guidelines: http://journals.plos.org/plosone/s/licenses-and-copyright.

a. You may seek permission from the original copyright holder of Figures 1 and 2 to publish the content specifically under the CC BY 4.0 license. 

a. If you are unable to obtain permission from the original copyright holder to publish these figures under the CC BY 4.0 license or if the copyright holder’s requirements are incompatible with the CC BY 4.0 license, please either i) remove the figure or ii) supply a replacement figure that complies with the CC BY 4.0 license. Please check copyright information on all replacement figures and update the figure caption with source information. If applicable, please specify in the figure caption text when a figure is similar but not identical to the original image and is therefore for illustrative purposes only.

Reviewers' comments:

Reviewer's Responses to Questions

**Comments to the Author**

1. Is the manuscript technically sound, and do the data support the conclusions?

Reviewer #1: Yes

Reviewer #2: Yes

2. Has the statistical analysis been performed appropriately and rigorously?

Reviewer #1: Yes

Reviewer #2: Yes

3. Have the authors made all data underlying the findings in their manuscript fully available?

Reviewer #1: Yes

Reviewer #2: Yes

4. Is the manuscript presented in an intelligible fashion and written in standard English?

Reviewer #1: Yes

Reviewer #2: Yes

5. Review Comments to the Author

Reviewer #1: Why just include men who are 50 years? Was it to just include men who are first invited by the programme to participate (i.e their first screening test). What about men aged >50 years?Other screening programmes have found age related differences in screening uptake

I may have missed this in the paper but are men screened on an annual basis? Or what is the screening interval?

Discussion: 'Comparisons with established cancer screening programmes' – maybe add that the relative newness of the the implementation of the OPE programmes may be a factor for lower participation compared to other programmes

Discussion: 'in Sweden, 293 neighbourhood income is more strongly associated with OPT participation than neighbourhood education level and proportion of non-Western immigrants in the neighbourhood' - can you give a possible rationale for this finding?

I think it is also important to point out that what might work in terms of interventions in one screening programme (e.g. cervical) may not work in other programmes (e.g. prostate) as you are dealing with different populations:'Observed socioeconomic inequality in screening programmes should lead to interventions

aimed at reducing disparity. While many such interventions have been explored, evidence for

their effectiveness remains limited (31, 32). Reminders and increasing accessibility, such as

302 providing home-based testing options, have proven effective in some screening contexts (7,

303 33, 34). Home-based PSA testing is being evaluated in some Swedish regional OPT

programs as a measure to increase participation rates, particularly for men in rural areas with a long distance to a testing facility.'

Reviewer #2: Row 86 –when possible morbidity

Please, provide the definition of deprivation related to your paper and also criteria for economic status in USD.

Part “Population distribution across neighbourhoods” from the Suppplement should be included in methodology but I suggest several changes:

Please provide more information about the importance of “country of birth” related to this sentence: "The dataset included 56,362 men aged 50-54 years, with information on disposable income and country of birth."

Please, provide the criteria for economic status in USD related to this sentence:"Economic status was categorized as low (disposable income below the lowest quartile of Swedish households), medium (middle two quartiles), and high (upper quartile)"

"By integrating this data with the DeSO dataset, we analysed the distribution of this population across the DeSOs in relation to the examined socioeconomic factors for each quintile and levels of urbanisation."Suggestion for changes: in relation to the examined socioeconomic factors according to each quintile and level of urbanization

Unclear context: "If the Region Västra Götaland population completely matched the national population, the valid percentage for each quintile would be 20%. 41,747 (74.1%) of the men were living in rural areas."

6. PLOS authors have the option to publish the peer review history of their article (what does this mean? ). If published, this will include your full peer review and any attached files.

**Do you want your identity to be public for this peer review?** For information about this choice, including consent withdrawal, please see our Privacy Policy .

Reviewer #1: **Yes: ** Dr Mairead O'Connor

Reviewer #2: No

---

## [Author Response · Author response to Decision Letter 1]

13 Mar 2025

All comments from editors and reviewers have been answered in the "Response to Reviewers" file uploaded in the file section.

---

## [Decision Letter · Decision Letter 1]

25 Mar 2025

Associations between neighbourhood characteristics and participation in a population-based organised prostate cancer testing (OPT) programme: a register-based study of 50-year-old men

PONE-D-25-02150R1

Dear Dr. Järbur,

We’re pleased to inform you that your manuscript has been judged scientifically suitable for publication and will be formally accepted for publication once it meets all outstanding technical requirements.

Kind regards,

Lorenzo Righi

Academic Editor

PLOS ONE

Additional Editor Comments (optional):

Reviewers' comments:

Reviewer's Responses to Questions

**Comments to the Author**

1. If the authors have adequately addressed your comments raised in a previous round of review and you feel that this manuscript is now acceptable for publication, you may indicate that here to bypass the “Comments to the Author” section, enter your conflict of interest statement in the “Confidential to Editor” section, and submit your "Accept" recommendation.

Reviewer #1: All comments have been addressed

Reviewer #2: All comments have been addressed

2. Is the manuscript technically sound, and do the data support the conclusions?

Reviewer #1: Yes

Reviewer #2: Yes

3. Has the statistical analysis been performed appropriately and rigorously?

Reviewer #1: Yes

Reviewer #2: Yes

4. Have the authors made all data underlying the findings in their manuscript fully available?

Reviewer #1: Yes

Reviewer #2: Yes

5. Is the manuscript presented in an intelligible fashion and written in standard English?

Reviewer #1: Yes

Reviewer #2: Yes

6. Review Comments to the Author

Reviewer #1: (No Response)

Reviewer #2: (No Response)

7. PLOS authors have the option to publish the peer review history of their article (what does this mean? ). If published, this will include your full peer review and any attached files.

**Do you want your identity to be public for this peer review?** For information about this choice, including consent withdrawal, please see our Privacy Policy .

Reviewer #1: No

Reviewer #2: No

---

## [Editor Report · Acceptance letter]

PONE-D-25-02150R1

PLOS ONE

Dear Dr. Järbur,

I'm pleased to inform you that your manuscript has been deemed suitable for publication in PLOS ONE. Congratulations! Your manuscript is now being handed over to our production team.

Kind regards,

on behalf of

Dr. Lorenzo Righi

Academic Editor

PLOS ONE